# In Vitro Antiplasmodial and Cytotoxic Activities of Compounds from the Roots of *Eriosema montanum Baker f.* (Fabaceae)

**DOI:** 10.3390/molecules26092795

**Published:** 2021-05-10

**Authors:** Jean Claude Didelot Tomani, Olivier Bonnet, Alain Nyirimigabo, William Deschamps, Alembert Tiabou Tchinda, Olivia Jansen, Allison Ledoux, Marie Jeanne Mukazayire, Luc Vanhamme, Michel Frédérich, Raymond Muganga, Jacob Souopgui

**Affiliations:** 1School of Medicine and Pharmacy, College of Medicine and Health Sciences, University of Rwanda, Kigali P.O. Box 3286, Rwanda; Jean.Claude.Tomani@ulb.be (J.C.D.T.); alain.nyirimigabo@yahoo.com (A.N.); jmukaza@yahoo.fr (M.J.M.); mugangaray@gmail.com (R.M.); 2Department of Molecular Biology, Institute for Molecular Biology and Medicine, Université Libre de Bruxelles, 6041 Gosselies, Belgium; William.Deschamps@ulb.be (W.D.); luc.vanhamme@ulb.be (L.V.); 3Centre for Interdisciplinary Research on Medicines (CIRM), Laboratory of Pharmacognosy, University of Liège, B36, 4000 Liège, Belgium; olivier.bonnet@uliege.be (O.B.); ojansen@ulg.ac.be (O.J.); allison.ledoux@uliege.be (A.L.); M.Frederich@uliege.be (M.F.); 4Laboratory of Phytochemistry, Centre for Research on Medicinal Plants and Traditional Medicine, Institute of Medical Research and Medicinal Plants Studies, Yaoundé P.O. Box 13033, Cameroon; talembert@gmail.com

**Keywords:** malaria, *Eriosema montanum*, isolation, pure compounds, antiplasmodial activity, cytotoxicity, THP-1

## Abstract

Malaria remains one of the leading causes of death in sub-Saharan Africa, ranked in the top three infectious diseases in the world. Plants of the *Eriosema* genus have been reported to be used for the treatment of this disease, but scientific evidence is still missing for some of them. In the present study, the in vitro antiplasmodial activity of the crude extract and compounds from *Eriosema montanum Baker f.* roots were tested against the 3D7 strain of *Plasmodium falciparum* and revealed using the SYBR Green, a DNA intercalating compound. The cytotoxicity effect of the compounds on a human cancer cell line (THP-1) was assessed to determine their selectivity index. It was found that the crude extract of the plant displayed a significant antiplasmodial activity with an IC_50_ (µg/mL) = 17.68 ± 4.030 and a cytotoxic activity with a CC_50_ (µg/mL) = 101.5 ± 12.6, corresponding to a selective antiplasmodial activity of 5.7. Bioactivity-guided isolation of the major compounds of the roots’ crude extract afforded seven compounds, including genistein, genistin and eucomic acid. Under our experimental conditions, using Artemisinin as a positive control, eucomic acid showed the best inhibitory activity against the *P. falciparum* 3D7, a well-known chloroquine-sensitive strain. The present results provide a referential basis to support the traditional use of *Eriosema* species in the treatment of malaria.

## 1. Introduction

Malaria is a human life-threatening disease provoked by several species of the protozoan parasite *Plasmodium* transmitted through female *Anopheles* mosquitoes. According to the recently published world malaria report, malaria remains an important parasitic disease affecting about 229 million patients globally [1]. Reported malaria-associated deaths are evaluated at 409,000 people globally, with children under the age of five years being the most affected. Almost all cases are reported from low and middle income countries (LMICs) of the subtropical regions, and Africa accounts for about 94% of the total burden, with around 215 million of cases and 384,000 malaria-associated deaths [1]. In light of these statistics, the high mortality rates of malaria and its associated economic burden are of great concern. Among the first-line treatment strategies, artemisinin-based combination therapies (ACTs) are recommended for uncomplicated cases of malaria, while artesunate monotherapy is directed to complicated and severe cases of malaria. Though these treatments have widely contributed to the decrease of malaria-associated deaths, increasing evidence suggest that their efficacy is declining due to the increase of resistant strains [1,2]. The increased incidence of drug-resistant parasites is a global concern for prophylaxis and treatment, as it leads to an increase in deaths [2]. Moreover, reported toxicities and adverse effects caused by conventional antimalarial agents limit their effectiveness in malaria therapy. Therefore, there remains a need to identify new antimalarial drugs. A reliable therapeutic source which remains poorly explored is the traditional pharmacopeia. *Eriosema montanum Baker f.* (Fabaceae) is a medicinal plant growing in many African countries. Its leaves and roots are traditionally used for the treatment of several diseases including conjunctivitis, snake-bites, cough and asthma [3,4]. The defatted ethanol extract of this plant has been reported to possess inhibitory activity against DNA and RNA viruses, including VIH, Semliki, etc. [4], and anti-inflammatory activity through a modulating activity of the classical complement [5]. Plants from the *Eriosema* genus including *Eriosema psoraleoides*, *Eriosema affine* and *Eriosema crinitum* have been reported to be traditionally used for the treatment of malaria in Angola [6,7,8,9]. Mindful of the leading role of natural products in the development of new drugs, especially antimalarial drugs, and knowing that different plant species from the *Eriosema* genus were previously reported as being traditionally used in the treatment of malaria, we sought to investigate the anti-plasmodial activity of the major compounds from *Eriosema montanum* roots. Our work showed the antimalarial activity of this medicinal plant for the first time and opened new features for drug development.

## 2. Results

### 2.1. Major Compounds Isolated and Identified from the E. montanum Roots

The roots of *E. montanum* were harvested from the Arboretum of Ruhande/Rwanda and were treated and extracted as previously described by Tomani and collaborators [10]. A total amount of 200 g of the powder was extracted with the mixture of methanol:dichloromethane (3:1) to give 27.75 g (13.9%) of crude extract. The described experimental approach allowed us to purify major constituents of the crude extract by preparative HPLC (Appendix A); and identified by 1D and 2D Nuclear Magnetic Resonance (NMR) and infrared spectroscopy (IR); and by comparison to the literature (Appendix A). The purified major constituents consisted of the following, as illustrated below (Figure 1): eucomic acid (structure **1**) [11], and five isoflavonoids derivatives, namely 7-*O*-glucopyranosyl-isoprunetin (structure **2**) [12], genistin (structure **3**) [13], malonyl genistin (structure **4**) [14]; isoprunetin (structure **5**) [15], Isoluteolin (structure **6**) [16], and genistein (structure **7**) [13]. The corresponding numbered structures are as follows:

### 2.2. Antiplasmodial and Cytotoxicity Activities of Eriosema Montanum Compounds

The half maximal inhibitory concentration (IC_50_) values obtained with the crude extract and various isolated compounds of Eriosema montanum against the 3D7 strain of *P. falciparum* are given in Table 1. Results are expressed as the mean ± standard deviation (S.D.) of the IC_50_ of two, three or four independent experiments performed on different days. The IC_50_ ranged from 0.042 for the compound with the structure number 5 to 17.68 µg/mL for the crude extract where isoprunetin and eucomic acid showed the best antiplasmodial activity (IC_50_ = 0.042 µg/mL and 0.057 µg/mL, respectively) with a high selectivity index (5264.3 and 6577 respectively). Amongst the isolated and identified compounds, genistin, malonylgenistin and genistein (compounds **3**, **4** and **7**, respectively) showed the lowest antiplasmodial activity (7.867, >10 and 7.736 µg/mL, respectively) and the most cytotoxic (IC_50_ of 168.6, 181.9 and 116.1 µg/mL, respectively) compared to other compounds.

## 3. Discussion

In this study, the majority of active isolated compounds are isoflavonoids. Isoflavonoids are a class of tricyclic natural products distributed in many plants, especially in the Papilionoideae subfamily of the Leguminosae [17]. They have been reported to mainly possess antioxidant, anti-inflammatory, antiparasitic and antimalarial properties [18,19,20]. The crude extract of *E. montanum* showed inhibition of *P. falciparum* parasites in vitro. It is well known that within a plant extract, there can be a myriad of compounds some of which are active individually or act synergistically to be effective in a given disease [10]. In most of the cases, major compounds are responsible for the activity of the extract though it is not always the case [10]. Hence, major compounds of the crude extract, as identified by the HPLC-DAD analysis (Appendix A), were isolated and tested against the 3D7 chloroquine-sensitive strains of *P. falciparum*. Results obtained suggest that the presence of a methoxy group on the aromatic ring (B or C) enhances the activity of the bearer isoflavonoid. This phenomenon can also be noticed in the patent WO 99/49862, where the presence of a methoxy group seemed also to enhance both antigiardial and antimalarial activities of formonetin, pseudobaptigenin and related derived isoflavones [20]. However, this effect decreased when the methoxy group is on both rings. In addition, it was reported in the literature that the presence of a methoxy group or any electron donating group enhances other biological activities of chalcones and flavonoids [21,22,23]. Though isoflavonoids seem to have an important share of the antiplasmodial activity of the plant, eucomic acid was the most active among all the tested compounds (Table 1). Eucomic acid is a derivative of malic acid, a compound reported to have antiplasmodial activity [23]. However, the IC_50_ seems to be enhanced by a phenyl group since its reported IC_50_ was superior to 10 µM [23]. Malic acid-derived polymers are also known to be a biocompatible material since their degradation leads to malic acid, which is not toxic [24].

Genistein, the compound with structure number **7** (Figure 1) was reported to possess an in vivo antimalarial activity, especially recommended for a combination therapy for malaria-induced splenic impairment [25]. In this study, however, genistein was less active with the lowest selectivity index among all the tested isolates. To investigate the potential of the isolates and the crude extract as lead compounds for the development of novel antimalarial drugs, their cytotoxicity was investigated against the THP-1 acute leukemia cell line, allowing the determination of their respective selectivity indexes (SI). The selectivity index (SI) was assessed as the ratio between the inhibitory activity against THP-1 human cells (CC_50_) and *P. falciparum* (IC_50_). It has been suggested that the SI > 10 indicate a favorable safety window between the effective concentration against the parasite and the toxic concentration to the human cell [26]. In general, the tested compounds exhibited an impressive SI of over 2000, with the exception of the crude extract and compounds **3**, **7** and Fr 3 (Table 1). Moreover, eucomic acid, the compound with the structure number **1,** was previously reported to be nontoxic to different cell lines with CC_50_ > 100 µM [27] and this was confirmed by the present study. Additionally, 7-*O*-glucopyranosyl-isoprunetin, genistin, malonylgenistin, isoprunetin and genistein (compounds **2**, **3, 4**, **5** and **7** respectively) have been reported to be less toxic to both normal and cancer cells [28,29,30], which was confirmed by the present work. Nevertheless, the crude extract was more toxic than the remaining tested compounds. This may be explained by the fact that the crude extract contains other toxic compounds, including the newly isolated Montachalcone A, a prenylated dihydrochalcone, as reported by Umereweneza and coauthors [13]. It could also be due to a combinatorial effect of its isoflavonoids since it has been reported that a combination of genistin, malonylgenistin, genistein together with acetylgenistin exerted potent toxic effects on HepG2 hepatocarcinoma cells, yet when taken individually, they were not toxic [31].

Regardless of the in vitro assay method, compounds are considered to be of interest, and worthy of further studies, when their IC_50_ ≤ 3.0 µM [32]. Considering the very high selectivity indexes for the eucomic acid, 7-*O*-glucopyranosyl-isoprunetin, isoprunetin and Isoluteolin (compounds **1**, **2**, **5** and **6**, respectively), these compounds are potential leads and worthy to be further investigated in vivo. If in vivo tests confirmed their efficiency, they would be potential candidates for galenic formulations.

## 4. Conclusions

The *Eriosema* genus is a source of compounds that can serve in the treatment of a wide range of ailments. Its traditional antimalarial potential has been reported in different communities. The in vitro antiplasmodial potential of isolated compounds from *Eriosema montanum*, reported in this study may explain the traditional use of several species of the genus in the treatment of malaria. Obviously, eucomic acid and isoflavonoids, at large, may contribute to the antimalarial activity of this plant. Therefore, our findings open new opportunities for drug development since this study showed for the first time the isolation of eucomic acid, 7-*O*-glucopyranosyl-isoprunetin, malonylgenistin, isoprunetin and isoluteolin from *Eriosema montanum* and it is the first to report the antiplasmodial activity of a plant from this genus.

## 5. Material and Methods

### 5.1. Collection of Plant Material *and Crude Extract Preparation*

Roots of *Eriosema montanum Baker f.* were collected from the arboretum of Ruhande, in Huye district, Southern province in Rwanda. The plant species was identified at the National Herbarium of Rwanda (University of Rwanda, Huye, Rwanda) and the authentication and ID verification were done at the Botanic Garden Meise (Meise, Belgium) as previously reported [3]. The voucher specimen (Tomani JCD 020) was deposited at both the National Herbarium of Rwanda and Botanic Garden Meise. The roots were cleaned of dust and debris by gently washing them with tap water. They were then chopped into small pieces, air-dried under shade, and thereafter pulverized with a grinder. A total amount of 200 g of the powder was extracted with the mixture of methanol:dichloromethane (3:1) as previously described [3].

### 5.2. Isolation and Structure Elucidation of Compounds

Major compounds from the crude extract of *E. montanum* were isolated by preparative HPLC. To this end, an in-house HPLC method used to screen the major components of crude extracts was used as previously reported [10]. Briefly, analytical separation was carried out on a Hypersil ODS^®^ RP18 column (250 × 4.6 mm; particle size 5 µm). All samples were dissolved in methanol HPLC-grade, filtered through a 0.45 μm pore size filter membrane and analyzed on an Agilent 1100 HPLC machine. Samples were eluted with a nonlinear gradient method with acetonitrile (solvent A) and 0.05% trifluoroacetic acid in ultra-pure water (solvent B) (Table 2). The column temperature was maintained at 25 °C. Then, 20 μL of each sample was injected into the HPLC-UV/DAD system and the analysis, performed at a flow rate of 1.0 mL/min, was monitored at 210, 254, 288, and 350 nm.

Preparative HPLC analysis was carried out on a Variant PrepStar machine. All extracts (500 mg) were dissolved in methanol and then diluted with water. The amount of methanol was not allowed to exceed 30%, to allow a good separation. Samples were filtered through a 0.45 µm filter membrane before injection. The mobile phase consisted of trifluoroacetic acid (TFA) 0.05% in ultrapure water (A) and acetonitrile (B). Table 3 gives the gradient used to separate the major compounds identified with the HPLC analytical method described above. This gradient was obtained by transposing the analytical HPLC methods to preparative HPLC using HPLC calculator. The flow rate was 15 mL/min and the separation was monitored at 254 and 350 nm. Structural identification of isolated compounds was performed by 1D and 2D nuclear magnetic resonance (NMR) and infrared spectroscopy (IR) and by comparison to the literature.

### 5.3. In Vitro Antiplasmodial Activities

Continuous in vitro cultures of asexual erythrocyte stages of *P. falciparum*, chloroquine-sensitive 3D7 strain was maintained according to the procedure of Trager and Jensen [33]. Strains were obtained from ATCC, Bei Resources. The host cells were human red blood cells (A+). The culture medium comprised RPMI 1640 (Gibco, Fisher Scientific, Merelbeke, Belgium) containing NaHCO_3_ (32 mM), HEPES (25 mM), and L-glutamine. The medium was supplemented with 1.76 g/L of glucose (Sigma-Aldrich), 44 mg/mL of hypoxanthine (Sigma-Aldrich, overijse, Belgium), 100 mg/L of gentamycin (Gibco, Fisher Scientific), and 10% human pooled serum (A+), as previously described [34]. Compound stock solutions were prepared in DMSO at 10 mg/mL (for an extract) and 1 mg/mL (for an isolated compound). Then, the solution is directly diluted in the medium; each test sample was applied in a series of eight 2-fold dilutions in a 96-well plate, starting with 100 µg/mL and 10 µg/mL for the crude extract and purified compounds, respectively, and tested in triplicate. The effect of compounds on the parasite growth was estimated, after 48 h incubation, by using the SYBR Green, a DNA intercalating compound. Artemisinin (Sigma-Aldrich, Machelen, Belgium), at an initial concentration of 100 ng/mL, was used as positive control in all experiments. The half-maximal inhibitory concentration (IC_50_) values were calculated from graphs.

### 5.4. Cytotoxicity Evaluation and Selectivity Index Determination

The cytotoxic effect of the crude extract and compounds thereof on THP-1 human acute leukemia cell line was carried using the Cell Titer-Glo Luminescent Cell Viability Assay (Promega). Briefly, cells were cultured in RPMI-1640 medium (Roswell Park Memorial Institute) supplemented with 10% heat-inactivated fetal bovine serum, 1% glutamine, and 1% of penicillin–streptomycin and maintained at 37 °C in a humidified 5% CO_2_ atmosphere. Cells were then incubated in the presence or absence of tested compounds for 3 days at 37 °C in a humidified 5% CO_2_ atmosphere in a 96-wells plate at a cell density of two thousand cells/well. The cell viability was then evaluated using Cell Titer-Glo Luminescent Cell Viability Assay (Promega) according to the manufacturer’s instructions. The 50% cytotoxic concentration (CC50) values were calculated from graphs, and the selectivity index was calculated as:

SI = CC_50_/IC_50_

## Figures and Tables

**Figure 1 molecules-26-02795-f001:**
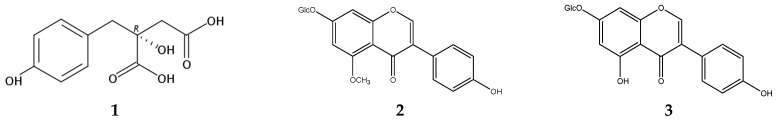
Structures of isolated and identified compounds from the roots of *E. montanum.* Preparative HPLC analysis was carried out on a Variant PrepStar machine. All extracts (500 mg) were dissolved in 30% methanol in distilled water. Samples were filtered through a 0.45 µm filter membrane before injection. The mobile phase consisted of trifluoroacetic acid (TFA) 0.05% in ultrapure water (solvent A) and acetonitrile (solvent B) as detailed in the Materials and Methods section. The gradient was obtained by transposing the analytical HPLC methods to preparative HPLC using HPLC calculator. The flow rate was 15 mL/min and the separation was monitored at 254 and 350 nm. Structural identification of isolated compounds was performed by 1D and 2D nuclear magnetic resonance (NMR) and infrared spectroscopy (IR) and by comparison to the literature.

**Table 1 molecules-26-02795-t001:** In vitro antiplasmodial activity of the crude extract and isolated compounds on the well-known 3D7 chloroquine-sensitive strain of *P. falciparum*, their THP-1 cell cytotoxicity and selectivity indices (SI).

Compound	Antiplasmodial Activity IC_50_ (µg/mL)	Cytotoxic Activity CC_50_ (µg/mL)	SI
** 1 **	0.057 ± 0.031	374.9 ± 97.3	6577
** 2 **	0.113 ± 0.074	321.7 ± 20.6	2846.9
** 3 **	7.867 ± 1.721	196.3 ± 44.5	24.9
** 4 **	>10	223.3 ± 58.6	n.d.
** 5 **	0.042 ± 0.028	221.1 ± 33.3	5264.3
** 6 **	0.121 ± 0.048	242.7 ± 73.0	2005.8
** 7 **	7.736 ± 0.802	116.1	15.0
** Fr-3 **	5.635 ± 0.6958	202.4	35.9
Crude extract	17.68 ± 4.030	101.5 ± 12.6	5.7
Artemisinin	0.0067 ± 0.003	n.d	n.d

Fr-3: Semi-purified fraction.

**Table 2 molecules-26-02795-t002:** Mobile phase gradient for the analytical high-performance liquid chromatography (HPLC).

Time (min)	Mobile Phase Proportion (%)	
	Solvent A	Solvent B
0.0	0.0	100
1.0	3.0	97.0
45.0	40.0	60.0
55.0	40.0	60.0
56.0	60.0	40.0
66.0	60.0	40.0
67.0	0.0	100
82.0	0.0	100

**Table 3 molecules-26-02795-t003:** Gradient system used to isolate compounds on preparative HPLC.

Time (min)	Mobile Phase Proportion (%)	
	Solvent A	Solvent B
0.0	0.0	100
2.0	3.0	97.0
35.0	15.5	84.5
55.0	19.0	81.0
76.0	27.5	72.5
98.0	31.5	68.5
133.0	60.0	40.0
144.0	60.0	40.0

## Data Availability

Please refer to suggested Data Availability Statements in section “MDPI Research Data Policies” at https://www.mdpi.com/ethics.

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
