# Peer review of "In Vitro Antiplasmodial and Cytotoxic Activities of Compounds from the Roots of Eriosema montanum Baker f. (Fabaceae)"

_molecules, 2021, doi:10.3390/molecules26092795_

Round 1

Reviewer 1 Report

Authors have already published several papers considering different plant species activity, evaluated by similar methodology. The paper is informative and presents the next step of the antimalarial activity recognition, belonging to different compounds in relation to their structure. Thus the value of these results is raised. 

Yes, I agree with the conclusion that the findings open new features for drug development since this study showed for the first time the isolation of compounds with structures number 1, 2 and 4-6 from Eriosema montanum and it is the first to report the antiplasmodial activity of a plant from this genus.

The minor remarks

The editing of Table 1 needs to be improved

Line 117 “The crude extract of E. montanum showed inhibition of P. falciparum parasites” Please explain in vitro or in vivo?

Line 145 “P. falciparum” please write in italic

The manuscript needs a careful overview for editing improvement, also in the references.

Author Response

See attached pdf file

Reviewer 2 Report

The manuscript entitled „In vitro antiplasmodial and cytotoxic activities of compounds 3 from the roots of Eriosema montanum Baker f. (Fabaceae)” by authors Tomani et al., is a concise, but very well conceptualized and written manuscript.

In the present study, the in vitro antiplasmodial activity of the crude extract of Eriosema montanum Baker f. and compounds from Eriosema montanum roots were tested against the 3D7 strain of Plasmodium falciparum. The cytotoxicity effect of the compounds against a human cancer cell line (THP-1) was assessed to determine their selectivity index.

Bioactivity-guided isolation of the major compounds of the roots crude extract afforded seven compounds amongst which genistein, genistin, and eucomic acid. In our experimental conditions, using Artemisinin as a positive control, eucomic acid showed the best inhibitory activity against this P. falciparum 3D7.

I advise to accept the manuscript with minor corrections:

  1. Line 71/72- missing verb (were), add the name of the author of the paper
  2. Line 168 In vitro- italic
  3. Line 142 Add ...acute leukemia cell line
  4. Check the text for typo errors

Author Response

See attached pdf file
